# Active Perception for Grasp Detection via Neural Graspness Field

**Haoxiang Ma**[1]    **Modi Shi**[1]    **Boyang Gao**[2]    **Di Huang**[1]*

[1]State Key Laboratory of Complex and Critical Software Environment,
School of Computer Science and Engineering, Beihang University, Beijing, China
[2]Geometry Robotics
{mahaoxiang822,modishi,dhuang}@buaa.edu.cn, boyang.gao@geometryrobot.com

## Abstract

This paper tackles the challenge of active perception for robotic grasp detection in cluttered environments. Incomplete 3D geometry information can negatively affect the performance of learning-based grasp detection methods, and scanning the scene from multiple views introduces significant time costs. To achieve reliable grasping performance with efficient camera movement, we propose an active grasp detection framework based on the Neural Graspness Field (NGF), which models the scene incrementally and facilitates next-best-view planning. Constructed in real-time as the camera moves, the NGF effectively models the grasp distribution in 3D space by rendering graspness predictions from each view. For next-best-view planning, we aim to reduce the uncertainty of the NGF through a graspness inconsistency-guided policy, selecting views based on discrepancies between NGF outputs and a pre-trained graspness network. Additionally, we present a neural graspness sampling method that decodes graspness values from the NGF to improve grasp pose detection results. Extensive experiments on the GraspNet-1Billion benchmark demonstrate significant performance improvements compared to previous works. Real-world experiments show that our method achieves a superior trade-off between grasping performance and time costs. Code is available at `https://github.com/mahaoxiang822/ActiveNGF`.

## 1   Introduction

Learning-based robotic grasp synthesis [24] has been explored to enable the manipulation of various objects across different grippers, sensors, and scenarios. The completeness and accuracy of the scene representation significantly influence the performance of these methods, as the geometric ambiguity can confuse the synthesis of grasp poses. To address this issue, previous works [4, 8, 20] have employed multi-view information to reconstruct different 3D representations. However, as the robot needs to move when observing from different views, scanning the entire scene to achieve complete coverage incurs a substantial time cost and is challenging to apply in real-world environments.

To enhance the efficiency of multi-view perception for robotic grasping, previous works have introduced active perception methods to select the Next-Best-View (NBV). Part of the methods [13, 7, 5] apply active 3D reconstruction techniques, treating grasp detection as a secondary task. However, active reconstruction tends to select viewpoints that cover more unobserved space, which may not be optimal for grasp detection, as different regions have varying relevance to grasping. Consequently, active perception strategies based on 3D reconstruction have limited performance in grasp detection tasks. Recently, ACE-NBV [32] combines grasp detection and NBV planning by specially designing

---

*Corresponding author.

38th Conference on Neural Information Processing Systems (NeurIPS 2024).

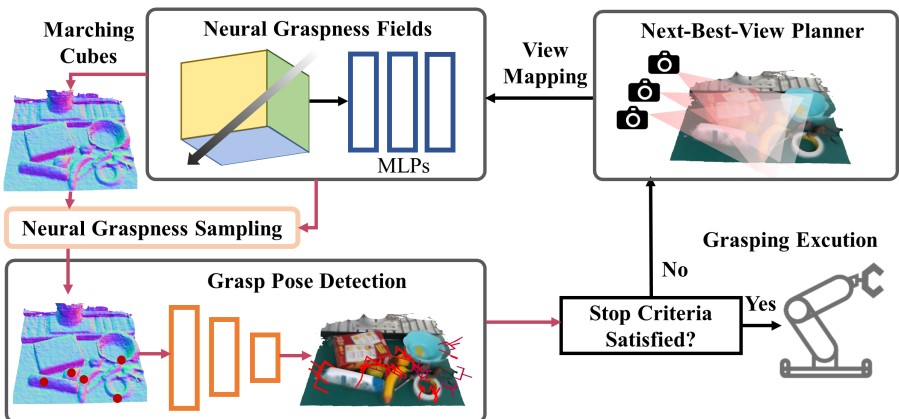

Figure 1: Overview of the active grasp detection system. The RGB, depth and predicted graspness from a new view are mapped to the Neural Graspness Field (NGF) by rendering loss. After each mapping step, the scene geometry is exported from the neural representation using the marching-cubes algorithm [19] and the candidate positions for grasp synthesis are sampled by neural graspness sampling. If the maximum perception step is reached or a specific result condition is satisfied, *e.g.*, a sufficient number of high-quality grasps are detected, the robot arm is employed to execute the detected grasps. Otherwise, the Next-Best-View (NBV) planner is employed to sample candidate views and select the view with the largest information gain for robot movement.

a grasp detection network to predict the grasp affordance of candidate views. The viewpoint with the highest predicted grasp quality is then selected for reconstruction to efficiently obtain feasible grasps. However, this approach has two main limitations. First, it suppresses the perception of views that have low grasp quality but provide grasp-related geometric information, which can lead the method to fall into local optima. Second, directly predicting information gain through a specially designed grasp detection network can be limited by the network's generalization ability, potentially resulting in performance degradation when applied to novel environments.

To address the aforementioned issues, inspired by the development of the neural representation and radiance field [21, 14], we propose to employ a Neural Graspness Field (NGF) to model the scene grasp distribution and build the active grasp detection system based on it. The graspness of a position measures the sum of feasible grasps in its pose space. The NGF is optimized online during the camera movement by back-propagating the rendering loss of the view graspness. The online-training scheme employed for NGF modeling makes it easier to transfer to unseen scenarios. Moreover, distilling multi-view information into 3D space using the neural representation enforces the multi-view consistency, making the NGF less susceptible to the depth noise and view sparsity compared to directly predicting the grasp distribution on a reconstructed 3D representation [33]. With the neural representation of the scene grasp distribution, the active perception problem can be defined as minimizing the error of the scene grasp distribution modeled by NGF. This is achieved by strategically selecting views that can bring the largest information gain for the NGF after mapping, thereby incrementally refining the modeled grasp distribution towards an optimal state.

In this paper, we propose an active perception method for the robotic grasp detection consisting of two components: neural graspness field mapping and graspness inconsistency-guided next-best-view planning. For neural graspness field mapping, we extend a NeRF-based real-time mapping system [12] to render view graspness by adding a separate network branch. The view graspness is predicted by a pre-trained graspness network from the corresponding depth image. For NBV planning, we provide a graspness inconsistency-guided strategy targeting minimizing the inconsistency between the current NGF and the ground-truth scene grasp distribution. For a given view, its information gain is described as the inconsistency between the view graspness rendered from the NGF and the pseudo label predicted from the rendered depth image. Furthermore, we propose an inference strategy based on our active grasp detection framework, which decodes the graspness score from the NGF to generate grasp samples instead of predicting from explicit 3D geometries. The contributions of this paper can be summarized as follows:

- We propose an active grasp detection framework via neural graspness field to model the scene grasp distribution online during camera movement.

- We adopt a graspness inconsistency-guided strategy for next-best-view planning, which targets on reducing the uncertainty of the neural graspness field.

- A neural graspness sampling inference strategy is proposed to enhance the performance of the grasp detection framework.

## 2 Related Work

### 2.1 Grasp Detection

Grasp detection aims to generate feasible and diverse gripper poses for given objects. Early methods mainly studied the theoretical framework for robotic grasping by analyzing the contacts between the gripper and object models. Recently, to achieve grasping of unseen objects, data-driven grasp detection methods have been extensively studied. Some works [9, 34, 1] investigate grasp detection in planar space, simplifying the problem by directly locating rotated grasp rectangles on images. To generate more diverse grasps that support complex downstream manipulation tasks, 6-DoF grasp detection has been proposed to predict grasping in $SE(3)$ space. [30] samples gripper poses on point clouds using geometric priors and scores these samples using a Convolutional Neural Network (CNN). [18] follows the sample-and-score framework but introduces PointNet [27] to achieve better scoring performance. Some recent works adopt end-to-end networks for 6-DoF grasp detection to achieve real-time inference. [23] employs a variational autoencoder for grasp generation given the object point clouds, and [28] proposes a single-shot grasp proposal network to enhance the efficiency of grasp detection. [10] densely annotates grasp labels in clutter to construct a large benchmark and provides an end-to-end 6-DoF grasp detection baseline trained with the large amount of annotations. Following this, [31] defines graspness to represent the grasp distribution in a scene and proposes a graspness discovery method. Although the aforementioned methods have achieved good results, using single-view point clouds as input leads to a lack of geometric information due to occlusions and limited field of view, which affects the grasping performance on some objects. To solve this problem, [4, 20] utilize Truncated Signed Distance Functions (TSDFs) to map multi-view frames before generating grasps, and [8] employs a generalizable neural radiance field to reconstruct the scene from cameras set around the scene, which supports the grasp detection of transparent and specular objects. However, these methods rely on scanning the scene from all possible views or several views surrounding the scene, resulting in excessive robot execution time for moving the camera. The large time cost reduces the usability of grasp detection methods in real-world scenarios.

### 2.2 Active perception for Grasp Detection

Active perception [3] aims to develop an agent that knows why it wishes to sense, and then chooses what to perceive, and determines how, when and where to achieve that perception. Planning the next-best-view is a commonly used method to realize an active perception system. In terms of grasp detection, active perception is introduced to determine the camera views that can achieve a trade-off between the grasp performance and time cost. There are two main lines of work in this area: those that treat grasp detection as a secondary task of 3D reconstruction and those that directly incorporate grasp detection into the view planning process. Some works [13, 2, 7, 5] plan the view sequence based on the 3D reconstruction metric of the grasp-relevant region, where the grasp detection is treated as a secondary task. [13] models the uncertainty of occluded voxels as a mixture of Gaussians and utilizes trajectory optimization to generate the view sequence. [2] provides an active vision approach to maximize surface reconstruction quality near the contact point region, and [7] adopts reinforcement learning for view planning with an object mask-guided reward function. To achieve close-loop control, [5] incorporates a grasp detection network to continuously predict grasps after each view mapping. However, employing grasp detection as a secondary task of 3D reconstruction overlooks the internal relation between grasp synthesis and scene reconstruction, which can lead to sub-optimal results. Another line of works [11, 22, 32] directly incorporates grasp detection into the view planning process. [11] maps grasp detection performance as a function of viewpoint for each object but struggles with novel objects. [22] uses the entropy of the network prediction to determine the next-best-view in a top-down grasping setting and recently, [32] proposes an affordance-driven policy based on an implicit grasp detection network to generate grasp affordance for unseen views. However, these approaches that utilize the output of grasp detection networks for view planning rely on the specific design of the network and are easily influenced by the generalization ability of the

grasp detection network, making it challenging to apply these methods to diverse real-world scenarios. In contrast to these approaches, our method utilizes view grasp distribution, which is independent of specific network designs, and constructs the NGF online. This enables our method to easily adapt to diverse real-world scenarios at test time.

## 3 Method

### 3.1 System Overview

The objective of grasp detection in cluttered scenes is to generate diverse feasible grasps for each object in the scene. Given a robotic arm with a mounted depth camera, active grasp detection aims to find a camera movement policy that can achieve high-quality results within a maximum of $T$ time steps. An overview of our active grasp detection system is provided in Fig. 1.

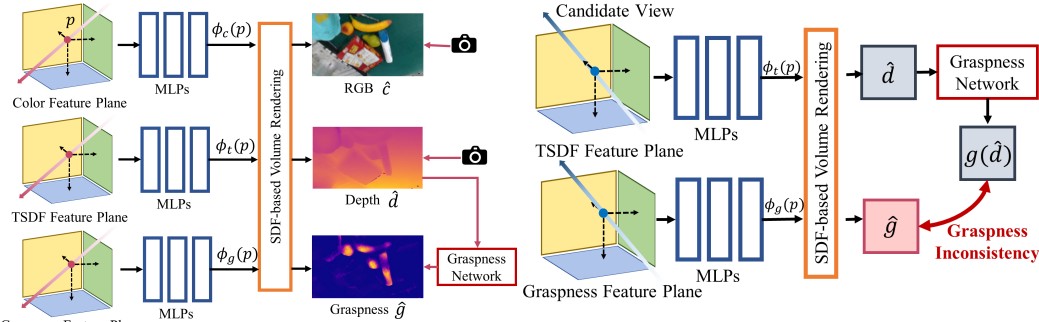

(a) Neural graspness Field Mapping      (b) Graspness Inconsistency-guided NBV Planning

Figure 2: The pipeline of the proposed mapping and NBV planning methods.

### 3.2 Neural Graspness Field Mapping

Inspired by neural feature fields [15] and semantic NeRF [33], we incorporate a graspness field to represent the scene grasp distribution. The NGF employs a separate branch besides the appearance and geometry to render grasp distribution information from multiple views. The graspness score proposed in [31] measures the graspable landscape in cluttered scenes given a position $p$. With $L$ grasp candidates $G_p = \{g_k^p | k = 1, ..., L\}$ sampled in its configuration space, *i.e.*, approach direction, gripper depth and in-plane rotation, the ground-truth graspness score $\widetilde{g}_p$ is defined as:

$$\widetilde{g}_p = \frac{\sum_{k=1}^{L} \mathbf{1}(q_k > t) \cdot \mathbf{1}(c_k)}{|G_p|} \tag{1}$$

where $q_k$ is the grasp quality score computed from force closure analysis, $c_k$ indicates the collision state of the gripper in clutter and $\mathbf{1}$ is the indicator function. For an observed view, the NGF aims to map the view appearance, depth and graspness, as shown in Fig. 2 (a). The NGF is composed of two parts: axis-aligned feature planes and SDF-based volume rendering. The axis-aligned feature planes store learned features at different resolutions, which are queried and interpolated based on the 3D positions of the sampled points. For points $p$ sampled in a ray, the corresponding features are queried from the feature planes and decoded by MLPs to get raw color $\phi_c(p)$, raw Truncated Signed Distance Field (TSDF) $\phi_t(p)$ and raw graspness $\phi_g(p)$. To convert raw TSDF values to volume densities, the SDF-based volume rendering from StyleSDF [25]:

$$\sigma(p) = \beta \cdot Sigmoid(-\beta \cdot \phi_t(p)) \tag{2}$$

where $\beta$ is a learnable parameter. With the volume density, the color, depth and graspness of each ray $r$ can be computed as:

$$\hat{c}(r) = -\sum_{n=1}^{N} w_n \phi_c(p_n) \quad \text{and} \quad \hat{d}(r) = -\sum_{n=1}^{N} \cdot z_n \quad \text{and} \quad \hat{g}(r) = -\sum_{n=1}^{N} w_n \phi_g(p_n) \tag{3}$$

where $z_n$ is the ray depth of $p_n$ and weight $w_n$ is formulated as:

$$w_n = exp(-\sum_{n-1}^{k=1} \sigma(p_k))(1 - exp(-\sigma(p_n))) \tag{4}$$

For a given view, it is challenging to directly obtain the corresponding ground-truth graspness during online mapping. Therefore, we use a pre-trained graspness network that takes the depth image as input and predicts the view graspness to render the NGF. By rendering from different views, the graspness field can reduce the graspness noise caused by single-view depth and render the graspness distribution from any given view, which can be used to measure the grasp-correlated information gain.

## 3.3 Graspness Inconsistency-guided Next-Best-View Planning

Given a sequence of viewpoints $s = (v_1, v_2, ..., v_N)$ and sequence set $S$, the NBV planning problem can be defined as:

$$s^* = \underset{s \in S}{argmax} \sum_{n=1}^{N} I(v_n) \tag{5}$$

where $N$ is the maximum step of robot movement and $I(v_n)$ represents the information gain of viewpoint $v_n$. The selection of an informative view is influenced by the scene representation and the definition of information gain. We consider the NGF obtained from observing the entire scene as the ground-truth scene grasp distribution and define the information gain of a view as the improvement in the NGF prediction by mapping this view. Given an unseen candidate view, the NGF can estimate the view graspness based on the existing observations, and the improvement is expressed as the inconsistency between the ground-truth graspness $g(d)$ predicted from the real depth and the graspness $\hat{g}$ rendered by the current graspness field.

However, for the NBV problem, obtaining the real depth image of a candidate view is not possible. To address this, our inconsistency-guided NBV policy adopts the pseudo-label paradigm by substituting the ground-truth grasp distribution $g(d)$ with the pseudo-graspness $g(\hat{d})$, which is widely employed in other semi-supervised and active learning vision tasks [17, 29]. As shown in Fig. 2 (b), we leverage the NGF's ability to render a pseudo-depth image $\hat{d}$ of the candidate view by volume rendering the TSDF values and then pass this pseudo-depth image through the graspness prediction network to obtain the pseudo-graspness $g(\hat{d})$. The pseudo-graspness is used to calculate the information gain of the candidate view, which is defined as:

$$I(v) = |\sum_{r \in v} \hat{g}(r) - g(\hat{d})| \tag{6}$$

where $g(r)$ represents the rendered graspness of sampled ray and the summation symbol represents the rendered graspness of the whole view.

It should be noted that the effectiveness of our NBV policy relies on the premise that the $g(\hat{d})$ predicted by the graspness network is closer to the ground-truth $g(d)$ compared to $\hat{g}$. The smaller error in $g(\hat{d})$ can be attributed to two reasons: First, the robot-mounted camera moves continuously in small steps, resulting in minimal differences in the observed geometric information between views, leading to insignificant errors in the rendered depth $\hat{d}$. Second, the graspness prediction network, trained on a large dataset of real point clouds, inherently provides robustness to depth noise. We visualize the rendered graspness error $E_{\hat{g}} = |g(d) - \hat{g}|$ and the pseudo-graspness error $E_{g(\hat{d})} = |g(d) - g(\hat{d})|$ in Fig. 3 (a), where $E_{\hat{g}}$ is significantly larger than $E_{g(\hat{d})}$ and the difference decreases with more steps.

The proposed pseudo-graspness information gain can incorporate the view grasp distribution prior into the planning process, which is encoded by the pre-trained graspness network. Thus the NBV system can select the view containing the most graspness information that has not been distilled from the pre-trained network to the NGF. We visualize the ground-truth graspness $g(d)$, pseudo-graspness $g(\hat{d})$, rendered graspness $\hat{g}$ and the information gain $I$ in different views of the neural graspness field in Fig. 3 (b). For different views, the pseudo-graspness predicted from the rendered depth image can approximately represent the ground-truth $g(d)$ but the accuracy of the rendered graspness $\hat{g}$ varies, which introduces different information gains.

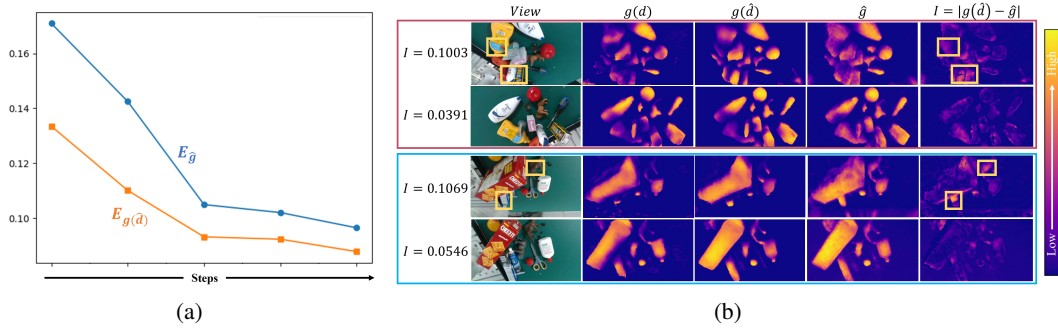

Figure 3: (a) The pseudo-graspness error $E_{g(\hat{d})}$ and rendered graspness error $E_{\hat{g}}$ of initial steps. (b) Visualization of the pseudo-graspness, rendered graspness and the corresponding information gain of different views.

## 3.4 Neural Graspness Sampling

For the grasp detection method, predicting the grasp distribution in clutter and sampling positions for grasp synthesis is an important part. Previous methods usually employ different encoders to predict grasp distribution from explicit 3D representations but the incomplete and noisy geometry information can lead to inaccurate grasp distribution. To achieve more precise grasp sampling, in addition to using the NGF for active perception, we propose an inference strategy based on sampling from the NGF. Given the positions $p$ sampled from the reconstructed surface, the graspness of the position can be decoded from the NGF. During inference, we replace the graspness sampled from the neural representation with the graspness predicted by the grasp detection network and utilize Furthest Point Sampling (FPS) on the positions larger than a threshold $T$ to get positions for grasp synthesis, which is formulated as:

$$Samples = FPS(p\{\phi_g(p) >= T\}) \tag{7}$$

where $\phi_g$ is the graspness branch of the NGF.

## 4 Experiments

### 4.1 Experimental Setup

**Simulation Setup** We construct a simulation active grasp benchmark based on the GraspNet-1Billion benchmark [10], which consists of 100 scenes for training and 90 scenes for testing. The test set is divided into seen, similar, and novel sets based on the included objects. Each scene is captured from 256 views using Intel RealSense and Kinect cameras. We conduct all the experiments with the data captured by the Realsense camera. We set the pre-collected 256 views as the perception space for NBV planning. Since moving the camera is a continuous process, moving it over long distances would waste the information captured during the movement. Therefore, we sample the candidate views from the current view with a relatively small step size. In our experiments, we set the step size to 10cm and set maximum step to 10. For evaluation, we follow the metric used in the GraspNet-1Billion benchmark, which simulate grasps with friction $\mu$ ranging from 0.2 to 1.2 with interval $\delta\mu = 0.2$. Following [20], we sample 5 grasps for each object in the scene to calculate the average precision **AP**. The training and evaluation of the simulation experiments are conducted on a single NVIDIA V100 GPU.

**Baselines** We compared the following baseline methods to validate the effectiveness of our proposed method. The baselines can be divided into two categories: NBV for robotic grasping [5, 32] and NBV planning based on NeRF [26, 16]. Close-loop NBV [5] utilizes ray casting to calculate the number of unobserved voxels of objects, which drives exploration targeting occluded object parts. ACE-NBV [32] incorporates grasp affordance prediction into NBV planning and selects the view with the largest grasp affordance as the next-best-view. ActiveNeRF [26] proposes an plug-in uncertainty estimation method for NeRF based on Bayesian estimation and Uncertainty-policy [16] computes the entropy of the weight distribution of each ray as the uncertainty.

**Implementation Details** For the mapping of NGF, the first view is trained for 100 iterations and following views are trained for 50 iterations. For each ray, 32 points are sampled for stratified sampling and 8 points for importance sampling. Only the coarse planes in [12] are employed for mapping. For NBV planning, we downsample the original image to $1/8$ to sample rays for graspness rendering to speed up the computation of view information gain. We utilize the first-stage network of GSNet [31] which predicts the graspness score for point-clouds as the graspness network in this paper. For the grasp detection network used for inference, we adopt the baseline method from [20] which uses the reconstructed scene geometry as input. For each scene, 1024 points are sampled from the NGF for grasp pose synthesis.

## 4.2 Simulation Experiments

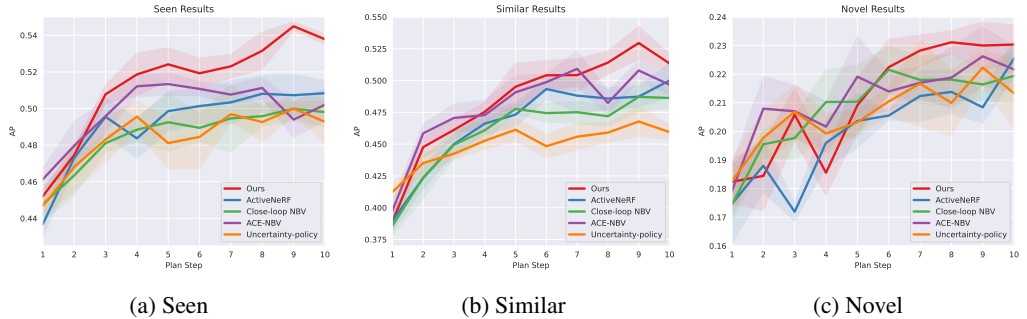

(a) Seen            (b) Similar            (c) Novel

Figure 4: Comparison on different NBV policies based on the proposed NGF.

**Comparison on different NBV policies** To validate the effectiveness of the proposed NGF and the graspness inconsistency-guided NBV policy based on it, we re-implement other view planning policies on the same ESLAM mapping framework [12] which the NGF is built on. As shown in Fig. 4, our pseudo-graspness guided policy achieves superior performance after the first several views on seen, similar, and novel sets. Compared to ActiveNeRF [26] and Uncertainty-policy [16], our method selects views with more grasp distribution discrepancy instead of geometry or appearance ambiguity in the neural representation, which improves the results. Compared to policies targeting grasp detection [5, 32], our method is specially designed to reduce the uncertainty of the NGF by distilling the prior knowledge of a pre-trained network, thus achieving superior performance with more views. It should be noted that ACE-NBV [32] can achieve comparable results to ours in the initial steps while showing little improvement as more views are added. This is because the affordance-based policy only selects views with more feasible grasps but does not consider optimizing the scene grasp representation throughout the entire planning process.

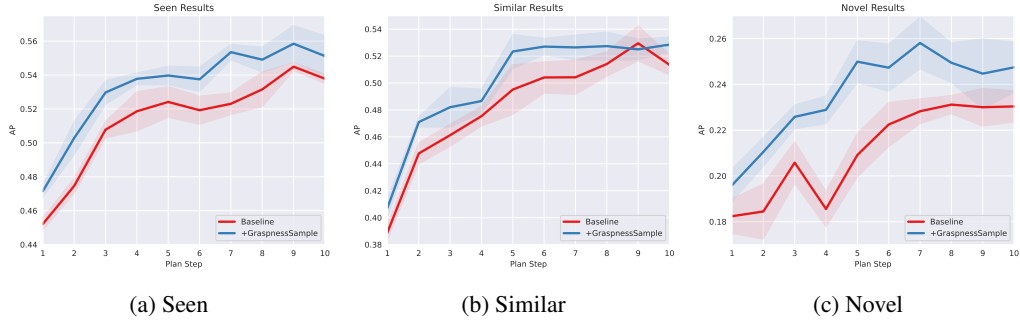

(a) Seen            (b) Similar            (c) Novel

Figure 5: The comparison of grasp detection result generated with the graspness predicted from 3D geometry and sampled from NGF.

**Effectiveness of Neural Graspness Sampling** We apply the neural graspness sampling during the inference of the grasp detection network to validate its effectiveness. The results are shown in Fig. 5. The sampling strategy improves the grasp detection results on seen, similar and novel objects at each step. Constructing a NGF through online multi-view rendering, compared to directly predicting

the grasp distribution from 3D geometric information using the network, can reduce the errors in the scene grasp distribution caused by incomplete geometric information. Furthermore, since the optimization is performed online for each scene, it demonstrates better robustness compared to direct network prediction and thus the results on the novel set improve significantly.

| Methods | Seen | | | Similar | | | Novel | | |
|---|---|---|---|---|---|---|---|---|---|
| | AP | $AP_{0.8}$ | $AP_{0.4}$ | AP | $AP_{0.8}$ | $AP_{0.4}$ | AP | $AP_{0.8}$ | $AP_{0.4}$ |
| Close-loop [5] | 43.84 | 53.95 | 34.18 | 42.17 | 51.51 | 34.02 | 19.54 | 23.96 | 9.49 |
| ACE-NBV [32] | 46.74 | 56.17 | 38.13 | 46.14 | 55.42 | 38.86 | 21.76 | 26.89 | 12.16 |
| Ours | 55.12 | 65.07 | 48.88 | 52.85 | 62.63 | 46.49 | 24.74 | 30.21 | 12.00 |
| All views | 63.75 | 73.30 | 58.38 | 61.54 | 71.17 | 55.94 | 24.89 | 30.18 | 13.95 |

Table 1: Overall results compared to the state-of-the-art active grasp detection methods.

**Overall Performance** We compare the overall performance after 10 views of our method with previous active grasp detection methods [5, 32] on the GraspNet-1Billion benchmark, as shown in Table 1. We employ the same grasp detection network trained on the GraspNet-1Billion benchmark for all these methods. Our active grasp detection method improves the performance by 8.38%, 6.71%, 2.98% on the seen, similar and novel sets compared to ACE-NBV [32] for grasp detection in clutter, demonstrating the effectiveness of the proposed method. All views represents a complete reconstruction using all 256 views, serving as an upper-bound reference for active perception methods.

### 4.3 Real-world Experiments

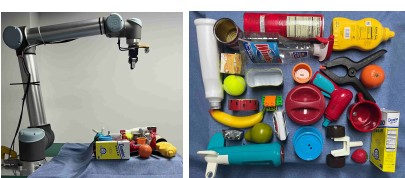

Figure 7: The robot setup of real-world experiments and the objects used for grasping.

| Model | Success Rate (%) |
|---|---|
| Close-loop [5] | 70.67 (53/75) |
| ACE-NBV [32] | 62.67 (47/75) |
| Ours | 74.67 (56/75) |

Table 2: Results of the real-world grasping experiments.

We conduct real-world experiments of the proposed active grasp detection method on a 6-DoF UR-10 robot arm with a mounted RealSense D435i depth camera. The robot setup and objects used for experiments are shown in Fig. 7. We select 25 objects from the YCB dataset [6] with various sizes and shapes for grasp detection. In the experiments, each cluttered scene is composed of 5 objects and we place these objects in different poses to evaluate each scene for 3 times. We employ the grasp success rate as the metric. As shown in Table 2, our method achieves 12.00% and 4.00% improvement on success rate compared to ACE-NBV [32] and Close-loop NBV [5], respectively.

| Overall | NBV Planning | Mapping | Grasp Detection | Robot Execution |
|---|---|---|---|---|
| 3.44s | 1.00s (29.07%) | 0.45s (13.08%) | 0.23s (6.69%) | 1.76s (51.16%) |

Table 3: Runtime analysis of the proposed method.

**Runtime Analysis** We provide a runtime analysis of the proposed active perception system, as shown in Table 3. The analysis is performed on a workstation with a single NVIDIA 3090 GPU and an AMD Ryzen 5 2600 six-core processor. The average execution time for each step is 3.44 seconds, with the robot execution accounting for approximately 50% of the total time. In the active grasp detection system, the majority of the time is consumed on NBV planning, while updating the NGF (mapping) and grasp detection take a relatively small proportion of the time. By investing some time in NBV planning, we achieve a trade-off between the performance and time cost compared to scanning the entire scene.

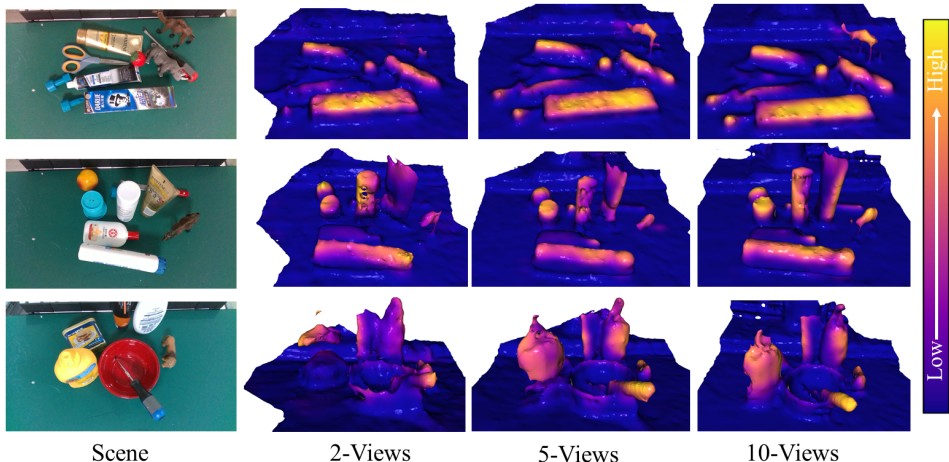

|   Scene   |   2-Views   |   5-Views   |   10-Views   |

Figure 8: Visualization of the geometry and graspness extracted from NGF in different planning steps.

## 4.4 Visualization of the Neural Graspness Field

Fig. 8 visualizes the NGF with different perception views, where the yellow region represents a higher grasp probability. It can be observed that the NGF can not only reconstruct the 3D geometry of the scene but also jointly model the graspness. With approximately 5 views planned using active grasp detection, the NGF can effectively model the grasp distribution of objects. As more steps are taken, the details of the geometry and grasp distribution of the scene can be incrementally refined.

## 4.5 Visualization of the Planned Camera Trajectories

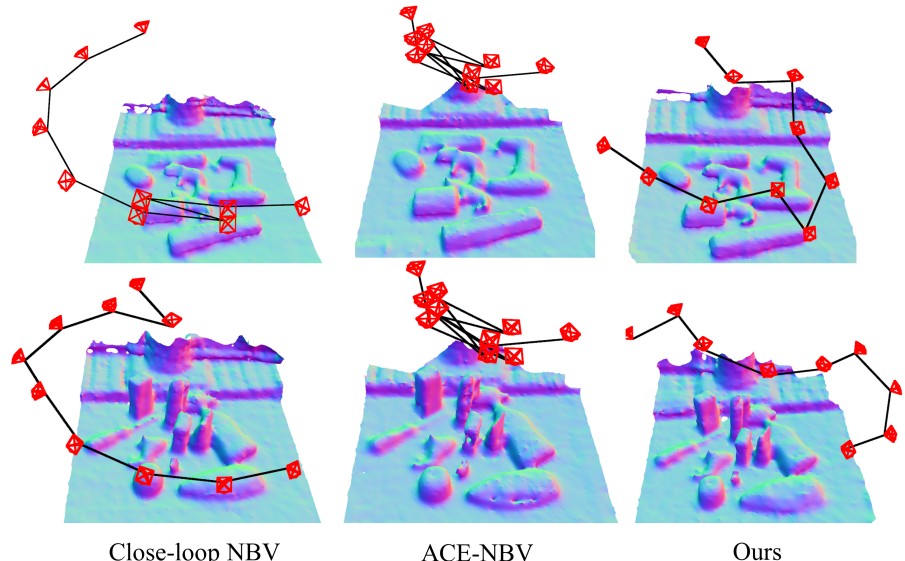

|   Close-loop NBV   |   ACE-NBV   |   Ours   |

Figure 9: Visualization of the camera trajectories generated from different active grasp detection methods.

Fig. 9 illustrates the view trajectories obtained by different view planning strategies. The close-loop NBV [5] approach, which employs unobserved space as the metric, guides the camera view path to scan regions with minimal overlap with the currently observed areas, aiming to maximize scene coverage. However, this method does not prioritize the graspable regions of the objects. In contrast, ACE-NBV [32] incorporates a grasp detection network to guide view planning by selecting views with the highest grasp affordance. Nevertheless, this approach tends to repeatedly scan a limited region, potentially leading to sub-optimal local results. Compared to these methods, our proposed

approach efficiently scans the grasp-correlated regions of the scene while ensuring comprehensive scene reconstruction.

## 5 Limitations

Our approach has two limitations. First, the time cost of NBV planning is positively correlated with the number of candidate views. When more refined view sampling is required, the planning time increases. Since our information gain computation is differentiable, this issue may be alleviated by sparse view sampling combined with pose optimization for the selected views. Second, our method cannot handle dynamic scene changes. Although efficient for static scenes, when grasping fails, *e.g.*, objects fall out of the gripper or change pose without being grasped, the robot must re-execute the perception process. Incorporating techniques used in dynamic radiance fields could potentially address this problem.

## 6 Conclusion

In this paper, we propose an active perception method for grasp detection by introducing the neural graspness field, which models the grasp distribution of a scene. By rendering the graspness predicted from a pre-trained network for each view, the NGF can be optimized online and reduce the noise of graspness in each view. Based on which, we introduce a graspness inconsistency-guided NBV policy to select the view with the largest inconsistency between the rendered graspness and pseudo-graspness label. Furthermore, we introduce neural graspness sampling to decode the grasp distribution from the neural representation, which benefits the position sampling of grasp pose synthesis. The experiments conducted on the simulation and real-world settings demonstrate the effectiveness of the proposed active grasp detection method.

## Acknowledgment

This work is partly supported by the National Natural Science Foundation of China (62022011), the Research Program of State Key Laboratory of Software Development Environment (SKLSDE-2023ZX-14), and the Fundamental Research Funds for the Central Universities.

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

# A   Appendix

| Methods | Seen | | | Similar | | | Novel | | |
|---|---|---|---|---|---|---|---|---|---|
| | AP | $AP_{0.8}$ | $AP_{0.4}$ | AP | $AP_{0.8}$ | $AP_{0.4}$ | AP | $AP_{0.8}$ | $AP_{0.4}$ |
| Close-loop [5] | 40.07 | 48.06 | 32.05 | 34.74 | 42.32 | 27.78 | 8.68 | 10.61 | 2.93 |
| ACE-NBV [32] | 44.74 | 54.23 | 36.32 | 37.72 | 45.69 | 31.65 | 13.56 | 16.51 | 7.46 |
| Ours | 52.35 | 61.64 | 45.86 | 44.50 | 51.76 | 39.82 | 13.94 | 18.02 | 5.97 |
| All views | 61.35 | 70.45 | 55.76 | 55.12 | 62.07 | 49.85 | 19.54 | 23.75 | 9.89 |

Table 4: Kinect results compared to the state-of-the-art active grasp detection methods.

**Overall performance on the kinect camera** We compare the overall performance of kinect camera on the GraspNet-1Billion benchmark, as shown in Table 4. Our active grasp detection method improves the performance by 7.61%, 6.78%, 0.38% on the seen, similar and novel sets compared to ACE-NBV [32] for grasp detection in clutter.

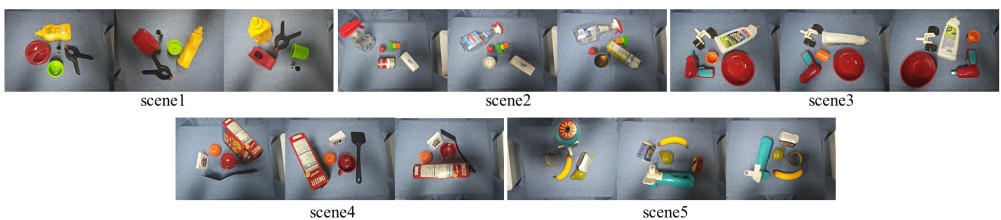

Figure 10: Object setting of the real-world experiment.

| Methods | Scene1 | | | Scene2 | | | Scene3 | | | Scene4 | | | Scene5 | | | Success Rate |
|---|---|---|---|---|---|---|---|---|---|---|---|---|---|---|---|---|
| | 1 | 2 | 3 | 1 | 2 | 3 | 1 | 2 | 3 | 1 | 2 | 3 | 1 | 2 | 3 | |
| Close-loop [5] | 4 | 4 | 3 | 3 | 5 | 3 | 4 | 4 | 3 | 4 | 3 | 3 | 4 | 3 | 3 | 53/75 |
| ACE-NBV [32] | 4 | 2 | 3 | 4 | 3 | 3 | 3 | 3 | 4 | 3 | 2 | 2 | 4 | 4 | 3 | 47/75 |
| Ours | 5 | 3 | 4 | 4 | 3 | 4 | 3 | 3 | 4 | 4 | 3 | 4 | 5 | 4 | 3 | 56/75 |

Table 5: Detailed results for each scene in real-world experiments.

**Details of the real-world experiment** The scene setting for real-world experiment is shown in Fig. 10. In total, we constructed 5 scenes, each containing 5 objects. For each scene, we repeated the experiment 3 times by changing the poses of the objects within the scene. The number of the success attempts for each scene is provided in Table 5.

