# OpenReview forum: "Active Perception for Grasp Detection via Neural Graspness Field"
_NeurIPS.cc/2024/Conference — NeurIPS 2024 poster_

### Official Review · Reviewer_DbvE · 2024-07-07

**Soundness:** 2
**Presentation:** 3
**Contribution:** 2
**Rating:** 5
**Confidence:** 4

**Summary:**

This paper proposes an next-best-view planning method for grasp perception. The authors use neural field to model the grasp distribution of a scene, which is learned from the graspness detection result of different views. Then the NBV is designed as the view with the largest inconsistency between the rendered graspness from the neural field and pseudo-graspness label predicted by the network on the rendered depth. Simulation and real-world experiements show that the proposed method is slightly bettern than exising methods.

**Strengths:**

The idea of modelling the graspness of the scene as a neural field is novel to my knowledge.
The paper is well structured and the visualization is good for readers. The method and the experiment detials are well presented and most parts are easy to understand and reproduce.

**Weaknesses:**

First of all, the meaning of NBV for grasp percetpion is not very clear to me. The goal of grasp percetpion is to find ONE successful and executable grasp in the scene, rather than detecing all the feasible grasps.

Secondly, the gain of the proposed method is not well validated by the experiments. In both simulation and real-world experiments, the proposed method is only slightly better than baselines. The margin is not significant.

Thirdly, the grasp network that is used to predict the graspness for neural field may produce inconsistent results across views. How do the authors address this problem during field optimization. The depth of the field for unseen views can also be errorneous.

**Questions:**

1. Sec 3.4 is not clear. Why do the authors need FPS? There is no ablation study on this design. Is the graspness score generated by the network or neural field?
2. Colorbar. should be added in Fig3, 8

3. How is the deviation between NGF and predicted graspness computed?

**Limitations:**

Yes, limitations are addressed in the paper: the time cost and the method cannot be used for dynamic scenes.

---

> ### Author Rebuttal · Authors · 2024-08-06
>
> **Response to Reviewer DbvE**
>
> Thanks for your valuable feedback. We understand your concerns about the goal of active grasp detection, the performance gain of our method and some other details. We address your concerns below:
>
> **Q1: Meaning of NBV for grasp perception.**
>
> While finding one feasible grasp pose allows a robot to pick up objects, grasp detection often serves as a part of semantically diverse downstream manipulation tasks [1, 2]. In such cases, it is crucial to provide diverse feasible grasp poses in the scene. For instance, grasping the cap area of a water bottle does not enable the pouring task. Similarly, when tidying up a table, the robot must select the grasping pose based on the object's placement pose. Therefore, it is reasonable for active perception to set the goal of grasp detection as finding all feasible grasps in the scene.
>
> **Q2: Performance gain of our method.**
>
> To evaluate the performance for active perception, we consider the grasp detection results with different perception steps, where our method totally show significantly superior results compared to other active perception method given same steps. As shown in Figure 4 of our original paper, our method consistently outperforms other NBV strategies by at least 1 AP starting from the 6th step across the seen, similar, and novel sets. In the ablation study of neural graspness sampling illustrated in Figure 5 of our original paper, the incorporation of this sampling method demonstrates significant improvements across almost all planning steps. Furthermore, Table 1 of our original paper presents the results of different active grasp detection methods under a 10-step planning scenario. Our method shows notable improvements compared to the previous state-of-the-art ACE-NBV, with increases of 8.38, 6.71, and 2.98 AP on the seen, similar, and novel sets, respectively. In the context of the Graspnet-1billion benchmark, these improvements can be considered substantial.
>
> **Q3: Problem about the process of NGF optimization.**
>
> In the optimization process of NGF, there indeed exists inconsistent graspness results for each observed view, which can be primarily controlled through two aspects: (1) During the optimization process, both the new and previously observed views are jointly involved. This approach fully utilizes multi-view information and promotes consistency in graspness across different views given the same spatial location. (2) According to previous research [3], neural representations inherently possess the capability to reconstruct relatively smooth distributions from sparse and noisy multi-view information.
>
> As for the depth error in the unseen views, becuase the camera moves continuously in small distance for each step, it is relative small in most cases and won't influence the calculation of the information gain. Figure1 in the rebuttal PDF also illustrate the effectiveness of using the rendered depth for the calculation of grasp information gain.
>
> **Q4: Explaination about Section3.4.**
>
> We apologize for any confusion. In Section 3.4, we introduce neural graspness sampling, which directly queries the graspness from NGF rather than predicting it from a grasp detection network. Following previous work [4], we incorporate FPS (Farthest Point Sampling) for two reasons: (1) To ensure that grasp positions are distributed throughout the entire scene. (2)To control the number of generated grasp poses, as there can be numerous positions with high graspness values.
>
> The graspness scores are generated by the neural field, and we generate grasp poses at positions where the graspness score exceeds a threshold.
>
> **Q5: Missing of the colorbar.**
>
> Thanks for your suggestion. In Figure 3 and Figure 8 of the original paper, blue means low graspness value while yellow represents high value. We will add it in the next version.
>
> **Q6: Deviation between NGF and predicted graspness.**
>
> During the training of NGF, the rendering process of graspness is similar to the color. We sample rays $r \in R$ from the given view and use L2 loss to calculate the deviation, which can be formulated as:
> $$
> L_g = \frac{1}{|R|}\sum_{r\in R}(\hat{g}(r)-g(r))^2
> $$
> where $\hat{g}(r)$ is the graspness rendered by NGF and $g(r)$ is the predicted graspness. During NBV planning, we employ Equation (6) in the original paper to calculate the deviation as the information gain.
>
> **References**
>
> [1] Learning task-oriented grasping for tool manipulation from simulated self-supervision, Fang et al., IJRR2020
>
> [2] Language Embedded Radiance Fields for Zero-Shot Task-Oriented Grasping, Rashid et al., CoRL 2023
>
> [3] In-Place Scene Labelling and Understanding with Implicit Scene Representation, Zhi et al., ICCV2021
>
> [4] Graspness discovery in clutters for fast and accurate grasp detection, Wang et al., ICCV2021

---

> > ### Comment · Reviewer_DbvE · 2024-08-12
> >
> > Thank you for the response. Most of my concerns are addressed, and I raise my rate to `borderline accept'

---

> > > ### Author Response · Authors · 2024-08-12
> > >
> > > We appreciate the reviewer for considering our response and are pleased that it has successfully addressed the concerns.

---

### Official Review · Reviewer_4AgP · 2024-07-09

**Soundness:** 3
**Presentation:** 3
**Contribution:** 3
**Rating:** 5
**Confidence:** 5

**Summary:**

This paper studies active perception for robotic grasp detection. It proposes an active grasp detection framework based on the Neural Graspness Field (NGF), which incrementally models the scene and facilitates next-best-view planning. For next-best-view planning, it aims to reduce the uncertainty of the NGF through a graspness inconsistency-guided policy, selecting views based on discrepancies between NGF outputs and a pre-trained graspness network. Additionally, it presents a neural graspness sampling method that decodes graspness values from the NGF to improve grasp pose detection results.

**Strengths:**

This paper considers how robotic arms can better select the next position to move towards the target object, which is rarely considered in current static two-finger grasping. I believe the authors' claim is important for the development of the robotics community. The authors have effectively combined NeRF with the significant concept of Graspness in recent years to study two-finger grasping, which I find to be an interesting topic.

**Weaknesses:**

Main concern：
1. First of all, I regret that the authors do not submit video materials in the supplementary materials. Although the authors analyze the real-world performance in the paper, the lack of videos makes it difficult to further verify the authenticity and effectiveness of the real-world experiments
2. The GraspNet-1Billion dataset includes two cameras, as mentioned in line L196 of the paper. However, in Table 1, the authors only provide experimental results for one camera, which I find insufficient. Testing on different cameras can more comprehensively demonstrate the effectiveness of the method.
3. Although the authors conduct sufficient comparison experiments to demonstrate the effectiveness of their method, I think they lack analysis experiments on some hyperparameters. For example, the authors set the maximum step to 10. I would like to ask the authors what the basis for setting it to 10 is, and what would happen if I used 7 steps or even fewer?
4. I think the recording of real-world experiments is insufficient. The authors should at least document the number of scenarios designed, the objects used in each scenario, and the average accuracy for each scenario. Simply recording an average success rate in Table 2 without detailing the experimental setup is, in my opinion, not rigorous. Additionally, there are no example images of cluttered scenes, making it difficult to assess the complexity of these scenarios. Typically, five objects do not constitute a cluttered scene; cluttered scenes usually consist of 10-15 or even more objects.
5. In section 4.5, I cannot intuitively see the clear performance advantages or disadvantages between Close-loop NBV and "Ours". In my opinion, the differences are due to the distinct training methods of the two planners, resulting in varied decision-making. I suggest the authors provide more distinguishable trajectory visualizations (e.g., whether the trajectory planning of Close-loop NBV causes collisions with objects, thereby disrupting the scene).

Others：
1. In equation (3), the formula for Pn​ is missing a closing parenthesis. Additionally, equation (3) does not clearly indicate the elements of ray tracing that are being focused on. For example, in NeRF, estimating density only requires the 3D location, while estimating color involves both the 3D location and the 2D viewing direction. The authors do not clearly explain in equation (3) which information c(r), d(r), and g(r) utilize.
2. In Table 1, the term "All views" is unclear, and there is no corresponding explanation provided in lines L245-L250.
3. Although the authors conduct an ablation study on GraspnessSample, I find the ablation experiments insufficient. As cited in the paper, the effectiveness of Graspness is already thoroughly validated in the GSNet paper. Conversely, the authors do not provide sufficient ablation studies on other parts of the planner. I believe these other parts are the more critical aspects of this paper.

**Questions:**

See Weaknesses.

**Limitations:**

The author mention the limitations of the paper and suggest possible solutions.

---

> ### Author Rebuttal · Authors · 2024-08-06
>
> #### **Response to Reviewer 4AgP**
>
> Thanks for your valuable feedback and we address your concerns below:
>
> **Q1: Video recording of real-world experiments.**
>
> Thanks for your suggestion. In **Figure 3** of the rebuttal PDF, we provide keyframe screenshots of one execution, including the active perception part and robotic grasping part. The videos will also be attached then.
>
> **Q2: Experiments on data captured from Kinect camera.**
>
> Thanks for your suggestion. We conduct additional experiments with the Kinect camera and make comparison with the previous SOTA method (i.e. ACE-NBV), as shown in **Figure 5** of the rebuttal PDF, demonstrating the effectiveness of our method under different cameras. Due to time and page limitation of the rebuttal, we only finish the experiments on the seen set and will supplement the results in the camer-ready version.
>
> **Q3: Analysis experiments on hyperparameters, such as the maximum NBV step.**
>
> In fact, we have presented the results with a varying number of steps for both the NBV policy comparison (Figure 4 in the original paper) and the Neural Graspness Sampling experiments (Figure 5 in the original paper). The horizontal axis denotes the number of planning steps. As the complete planning policy greedily chooses the view yielding the highest information gain at each step, the results with 7 or fewer steps can be directly inferred from the graph. Regarding the other hyperparameters, such as the number of iterations for NGF, we emprically know that they do not substantially influence the performance of our method. Anyway, we will improve our experiments on hyperparameter sensitivity in the next version.
>
> **Q4: More detials about the real-world experiment setting.**
>
> In this setting, we consider 5 scenes, each containing 5 objects, and for each scene, we repeat the experiment 3 times by changing the poses of the objects within the scene. We provide the detail recording of the real-world experiment in the table below.
>
> | Scene_id-Pose_id | 1-1  | 1-2  | 1-3  | 2-1  | 2-2  | 2-3  | 3-1  | 3-2  | 3-3  | 4-1  | 4-2  | 4-3  | 5-1  | 5-2  | 5-3  | Overall |
> | ---------------- | ---- | ---- | ---- | ---- | ---- | ---- | ---- | ---- | ---- | ---- | ---- | ---- | ---- | ---- | ---- | ------- |
> | Close-loop       | 4  | 4  | 3  | 3  | 5  | 3  | 4  | 4  | 3  | 4  | 3  | 3  | 4  | 3  | 3  | 53/75   |
> | ACE-NBV          | 4  | 2  | 3  | 4  | 3  | 3  | 3  | 3  | 4  | 3  | 2  | 2  | 4  | 4  | 3  | 47/75   |
> | Ours             | 5  | 3  | 4  | 4  | 3  | 4  | 3  | 3  | 4  | 4  | 3  | 4  | 5  | 4  | 3  | 56/75   |
>
> Although we use a relatively small number of objects (5) to construct each scene, the objects in the scene are arranged in a crowded manner, with the presence of view occlusions. Therefore, our scene setting reflects the characteristics of cluttered scenes. In the rebuttal PDF, we provide the object setting of the real-world experiment scenarios in **Figure 4**.
>
> **Q5: Differences between the trajectories generated by Close-loop NBV and ours .**
>
> Close-loop NBV is a heuristic active perception method based on TSDF Fusion that calculates the number of potentially unobserved voxels as information gain without involving a training process. However, this type of methods do not directly relate to grasp detection, as the view selection primarily depends on the overlap between the candidate views and the observed region. Therefore, the trajectory generated by the Close-loop NBV planner tends to uniformly surround the scene to maximize the observation area. More examples are shown in **Figure 6** of the rebuttal PDF.
>
> **Q6: Problem about Equation (3).**
>
> We apologize for the missing closing parenthesis and any confusion caused by Equation (3). Regarding the calculation for ray color $\hat{c}(r)$, depth $\hat{d}(r)$, and graspness $\hat{g}(r)$, the NGF follows the previous NeRF-SLAM mapping method [1], where only the 3D location is involved. However, unlike the vanilla NeRF, which directly decodes the color and depth from a single MLP, the mapping framework that we employ first queries the position-corresponding feature from the learnable axis-aligned feature planes and uses separate MLPs to decode the color, depth, and graspness. This approach can significantly enhance the convergence speed. Although not introducing view directions when rendering RGB may lead to a slight decrease in quality, we typically only focus on depth and graspness for grasp detection. Therefore, the mapping system based on the NeRF-SLAM framework is suitable for NGF.
>
> **Q7: Meaning of "All views" in Table 1.**
>
> In Table 1 of the original paper, "All views" refers to using all the 256 views captured for a scene in the Graspnet-1Billion dataset to perform a complete reconstruction of the scene and then infer the grasping results. This result serves as an upper-bound reference for active perception methods.
>
> **Q8: Ablation study about Neural Graspness Sampling.**
>
> We would like to clarify although the definition of graspness is proposed in GSNet, our method significantly differs in the way of graspness generation and achieves better results. In GSNet, a grasp detection network is used to infer the graspness value of each point from the input point cloud, which depends on the quality of the point cloud and does not effectively utilize multi-view information. Our proposed neural graspness sampling directly queries the graspness value by the position from the NGF, which fully leverages the NGF's capability to accurately model the scene's grasp distribution, leading to improved results. Therefore, it is essential to conduct an ablation study on neural graspness sampling to validate its contribution. For other parts of the NBV planner, we will strive to provide ablation studies in the next version.
>
> **References**
>
> [1] ESLAM: Efficient Dense SLAM System Based on Hybrid Representation of Signed Distance Fields, Johari et al., CVPR 2023

---

> > ### Comment · Reviewer_4AgP · 2024-08-10
> > **Official Comment by Reviewer 4AgP**
> >
> > Thanks to the author's reply, your experiments address most of my concerns, so I'm willing to raise my score to "borderline accept". Hopefully, you will be able to refine your ablation studies in the next version as mentioned in Q8's reply.

---

> > > ### Author Response · Authors · 2024-08-11
> > >
> > > We appreciate the reviewer's feedback and are pleased that our response has addressed the concerns, resulting in a raised score. We will refine the ablation study and finish the Kinect camera experiment to refine our paper.

---

### Official Review · Reviewer_peVt · 2024-07-11

**Soundness:** 2
**Presentation:** 2
**Contribution:** 2
**Rating:** 5
**Confidence:** 4

**Summary:**

The paper introduces a novel framework utilizing a Neural Graspness Field (NGF) in conjunction with a pre-trained graspness prediction network to enhance active grasp detection. It applies online training to the NGF upon encountering a new scene view, producing RGB, depth, and graspness maps. The method involves computing information gains from potential views by assessing discrepancies between the NGF-generated and pre-trained network graspness maps to select the most informative views. During grasp inference, the framework samples grasp poses guided by NGF-derived graspness scores and samples them using the Farthest Point Sampling (FPS) method.

**Strengths:**

1. The approach of leveraging graspness inconsistency to define information gain offers a targeted advancement for the robotic grasping task, potentially surpassing traditional methods that focus on uncovering occluded regions, as referenced in [1].
2. Comprehensive evaluations are conducted through both simulated and real-world experiments, with comparisons against multiple baselines. These experiments demonstrate the framework's effectiveness in practical scenarios.


[1] Breyer, Michel, et al. "Closed-loop next-best-view planning for target-driven grasping." 2022 IEEE/RSJ International Conference on Intelligent Robots and Systems (IROS). IEEE, 2022.

**Weaknesses:**

1. A primary concern is the incremental training approach of the NGF. Since the NGF is trained incrementally, the rendered graspness maps \hat{g} and depth maps might be inaccurate for novel views at the initial stage, and thus are inappropriate to be used to compute information gain.
2. The NGF's grasp knowledge, being distilled from the pretrained graspness network, suggests that its maximum achievable performance may be inherently limited to that of the teacher network. How can you demonstrate that the proposed framework is better than the single-shot method?
3. According to Eqn(1), the graspness score of a position is defined as the mean grasp quality scores of different orientations on that position. When sampling grasp poses at the inference stage, it seems that they only sample positions for the grippers. How to determine the orientations?
4. Visual results (Figure 3) indicate that initial high information gains may be attributed more to background inconsistencies rather than the target object regions, potentially skewing the focus of planning efforts. The inconsistency of the background region should be neglected when planning.

**Questions:**

1. Despite the reported rapid mapping operation time of 0.45 seconds in Table 3, the paper notes extensive training iterations for both initial and subsequent views (100 and 50 iterations). Further details on achieving such fast training times would be beneficial for understanding the feasibility of the NGF training process.

2. Equation (6) introduces confusion regarding the role of the summation symbol and the meaning of the variable $r$. Clarification of these components would aid in a better comprehension of the equation's intent and application.

**Limitations:**

See weaknesses and questions.

---

> ### Author Rebuttal · Authors · 2024-08-06
>
> #### **Response to Reviewer peVt**
>
> Thanks for your valuable feedback. We understand your concerns regarding the computation of information gain with NGP and the other issues you have raised. We address your concerns below:
>
> **Q1: Incremental training approach of the NGF.**
>
> In the initial stage of NGF, there are indeed errors in both the rendered depth $\hat{d}$  and graspness $\hat{g}$ but have few influence on the calculation of information gain.  Our inconsistency-guided NBV policy adopts the pseudo-label paradigm by substituting the ground-truth grasp distribution $g(d)$ with the pseudo-graspness $g(\hat{d})$, which is widely employed in other semi-supervised and active learning vision tasks [1,2]. The effectiveness of our NBV policy relies on the premise that the $g(\hat{d})$ predicted by the graspness network is closer to the ground-truth $g(d)$ compared to $\hat{g}$. The smaller error in $g(\hat{d})$ can be attributed to two factors: (1) The robot-mounted camera moves continuously in small steps, resulting in minimal differences in the observed geometric information between views, leading to insignificant errors in the rendered depth $\hat{d}$. (2) The graspness prediction network, trained on a large dataset of real point clouds, inherently provides robustness to depth noise. As a supplementary, we visualize the rendered graspness error $E_\hat{g} = |g(d)-\hat{g}|$ and the pseudo-graspness error $E_{g(\hat{d})} = |g(d)-g(\hat{d})|$ in **Figure 1** of the rebuttal PDF, where $E_\hat{g}$ is significantly larger than $E_{g(\hat{d})}$ and the difference decreases with more steps.
>
> **Q2: The maximum achievable performance of our method.**
>
> We are not certain the definition of "single-shot":
>
> If "single-shot" refers to using a single-view depth map for grasp detection, our active perception method aims to reconstruct the scene geometry and grasping representation with as few steps as possible. The scene point-cloud reconstructed by active perception is used for final grasp poses prediction,, resulting in significantly better performance.
>
> If "single-shot" refers to not using neural graspness sampling,  it's important to note that while the grasping knowledge of NGF originates from the teacher network's output, NGF enhances the multi-view consistency and smoothness of the graspness distribution through implicit multi-view fusion. This is particularly beneficial when the single-view output is noisy and views are sparse, as demonstrated in similar scenarios such as 3D segmentation [3]. Therefore, sampling grasping positions from NGF yields better results compared to directly sampling from the network's graspness prediction.
>
> **Q3: The inference pipeline with neural graspness sampling.**
>
> We are sorry for the confusion caused. To clarify, we still rely on a grasp detection network to perform inference on the reconstructed scene point-cloud after active perception, rather than directly sampling grasp poses from NGF. For neural graspness sampling, we replace the graspness output of the grasp detection network with values obtained from NGF based on the position, while parameters such as rotation and gripper width are still inferred using the grasp detection network.
>
> **Q4:  Visualization in Figure 3 in original paper.**
>
> In robotic grasping scenarios, it is difficult to divide the objects and background in 3D space from a single view perception. Therefore, given a candidate view, the graspness values of both foreground and background regions are used to compute the information gain. For our methods, not all selected views with high information gain arise from background errors. We provide more visualization results in the **Figure 2** of rebuttal PDF for illustration, where we use yellow rectangle box to highlight the forground inconsistency region.
>
> **Q5: The mapping time for our framework.**
>
> The mapping method is an extension of an efficient dense SLAM method [4] (citation [12] in our paper) that exploits axis-aligned feature planes and implicit Truncated Signed Distance Fields to reduce the number of training iterations, whereas previous dense neural mapping methods require more iterations for convergence.
>
> **Q6: Explaination about Equation(6).**
>
> We are sorry for the confusion caused. $g(r)$ means the rendered graspness of sampled ray $r$ and the summation symbol means the rendered graspness of the whole view. The definition follows Equation (3) in our paper and we will clarify it in the camera-ready version.
>
> **References**
>
> [1] Rethinking pseudo labels for semi-supervised object detection, Li et al., AAAI2022.
>
> [2] Learning From Synthetic Images via Active Pseudo-Labeling, Song et al., TIP2020
>
> [3] In-Place Scene Labelling and Understanding with Implicit Scene Representation, Zhi et al., ICCV2021.
>
> [4] ESLAM: efficient dense SLAM system based on hybrid representation of signed distance fields, Johari et al., CVPR2023.

---

> ### Comment · Reviewer_peVt · 2024-08-11
>
> I appreciate the authors' response, which addresses most of my concerns and helps me understand this work better. Here are some of my follow-up questions:
>
> 1. Since you already have the observed depth map, why do you use $g(\hat{d})$ to guide the training of $\phi_g(p)$ rather than $g(d)$?
>
> 2. For Q2, 'single-shot' refers to the first case mentioned above by the authors. It seems the single-shot method [1] already achieves better APs compared to the results reported in Table 1.
> Could the authors elaborate on the advantages of the proposed method compared to [1]?
>
> [1] Wang, Chenxi, et al. "Graspness discovery in clutters for fast and accurate grasp detection." Proceedings of the IEEE/CVF International Conference on Computer Vision. 2021.

---

> > ### Author Response · Authors · 2024-08-12
> >
> > Thanks for your feedback and we are pleased that our response addresses most of your concerns. As for the follow-up questions, we address them below:
> >
> > **Q1**: We use $g(d)$ to guide the training of $\phi_g(p)$ because we observed that after mapping a view, the view-rendered depth $\hat{d}$ has better quality than the original depth map observed by the camera, with fewer missing points and smoother surfaces. Using the rendered depth as input to calculate view graspness achieves better results than directly using the original depth. Based on this observation, we employ a two-stage training process. First, we optimize the RGB and depth. Next, we use the rendered depth to calculate $g(\hat{d})$ for the guidance of graspness.
> >
> > **Q2**: We understand your concern about the final performance, where our results reported in the paper are lower than GSNet, which only utilizes single view point-cloud as input.
> >
> > Firstly, we want to clarify that the target of GSNet and our method is quite different. Our method aims to build a complete 3D representation for grasp detection rather than design a grasp detection network, which means our method can be applied to any grasp detection method with 3D information as input. With actively reconstructed 3D information, grasp detection methods can achieve better performance especially for the unseen objects whose shape deviate from those in the training set. Unseen objects are not incorporated in the training process so the grasp detection netowrk struggles with the single-view input, where the geometry information is incomplete and noisy. Besides, utilizing multi-view information for 3D reconstruction can make the grasp detection method work on transparent and specular objects [1], while the single view depth map is in very low quality.
> >
> > For the final result, in our original paper, we use the baseline grasp detection network from [2] rather than GSNet for evaluation. For a fair comparison, we provide the results inferred by pre-trained GSNet here. CD stands for collision detection. With the input reconstructed by 10 steps of active perception, GSNet performs better on similar and novel sets while showing a performance drop on seen objects. Our reconstructed data is not involved during training, so we tend to attribute the performance drop on seen objects to the distribution gap between reconstructed and single-view point clouds. We will attempt to include the reconstructed point clouds for training to achieve better results in the future.
> >
> > | Method                | Seen-AP   | Similar-AP | Novel-AP  |
> > | --------------------- | --------- | ---------- | --------- |
> > | GSNet-singleview      | **65.70** | 53.75      | 23.98     |
> > | GSNet-10 steps        | 57.62     | **55.42**  | **24.53** |
> > | GSNet-singleview + CD | **67.12** | 54.81      | 24.31     |
> > | GSNet-10 steps + CD   | 61.78     | **61.60**  | **26.55** |
> >
> >
> >
> > **References**
> >
> > [1] Graspnerf: Multiview-based 6-dof grasp detection for transparent and specular objects using generalizable nerf, Dai et al., In *IEEE International Conference on Robotics and Automation*, 2023.
> >
> > [2] Generalizing 6-dof grasp detection via domain prior knowledge, Ma et al., In *IEEE/CVF Conference on Computer Vision and Pattern Recognition*, 2024.

---

> > > ### Author Response · Authors · 2024-08-13
> > >
> > > Dear Reviewer peVt,
> > >
> > > We appreciate your previous feedback and have provided our response to your questions. As the discussion phase is nearing its end on August 13, we want to check if you have had the opportunity to review our latest response.
> > >
> > > If you have any further concern or require additional clarification, we are willing to address them within the remaining discussion period.
> > >
> > > Thank you for your time and consideration.

---

> > > > ### Comment · Reviewer_peVt · 2024-08-14
> > > >
> > > > Thank you for your response. I think it answered my question. Therefore, I am willing to raise my score to borderline accept.

---

> > > > > ### Author Response · Authors · 2024-08-14
> > > > >
> > > > > Thanks for your feedback and we are pleased that our response has successfully addressed your concerns.

---

### Official Review · Reviewer_DQft · 2024-07-12

**Soundness:** 3
**Presentation:** 3
**Contribution:** 3
**Rating:** 7
**Confidence:** 3

**Summary:**

This work proposes an active perception method for grasp detection composed of two parts: neural graspness field mapping and next-best-view planning with a graspness inconsistency-guided strategy. And a corresponding inference strategy is also proposed by decoding the graspness score from NGF to generate grasp samples. The evaluation benchmark is constructed based on the GraspNet-1B benchmark and the experiment results indicate consistent improvements on the seen, similar and novel sets.

**Strengths:**

1. This work is well motivated: aiming to settle both the negative effect of incomplete 3D geometry information for learning-based grasp detection methods and the time costs of scanning the whole scene, and find a trade-off.
2. Experiments on GraspNet-1B demonstrate performance improvements compared to previous methods.

**Weaknesses:**

1. Relatively small performance improvement on the novel set compared to ACE-NBV.
2. Generalization and distraction experiments are not provided to verify model's robustness.

**Questions:**

1. Could you account for why with plan step increases, for the novel set, your method encounters a significant drop in performance?
2. Could you provide further generalization and distraction experiments to verify your model's robustness?

**Limitations:**

1. The NGF-based method may be constrained to static environments and cannot adapt to dynamic scenarios.

---

> ### Author Rebuttal · Authors · 2024-08-06
>
> **Response to Reviewer DQft**
>
> Thanks for your valuable feedback. We appreciate your acknowledgment of our active perception method based on neural graspness field and the performance improvements achieved. We address your concerns below:
>
> **Q1: The relative weak improvement on novel set.**
>
> In our experiments, the improvement of the proposed method compared to other NBV strategies is less pronounced on the novel set than on the seen and similar sets. We attribute this to several factors: (1) The grasp detection model for inference demonstrates relatively lower performance on the novel set compared to the seen and similar sets. This inherent limitation constrains the potential improvements through active perception. (2) The graspness prediction network we employed has not been exposed to objects from the novel set during training. Consequently, the view graspness prediction used for training the NGF may lack accuracy for novel objects. This potential inaccuracy in graspness prediction may impact the performance of our active perception approach.
>
> Our method also experiences a performance drop between step 3 and 4 in Figure 4 (c) of the original paper. We attribute this primarily to the grasp detection model's suboptimal performance on the novel set, which leads to a scarcity of positive grasp samples and makes the final results unstable.  As the number of view gradually increases, our active perception method still achieves stable performance improvement on novel set.
>
> **Q2: Generalization and distraction experiments.**
>
> Thanks for your suggestion. In future work, we will strive to supplement our study with ablation experiments on different modules and hyperparameters of our method to demonstrate its robustness and reliability.

---

> > ### Comment · Reviewer_DQft · 2024-08-12
> >
> > Thank you for your response to my concerns. I will keep my score and hopefully, you can add generalization and distraction experiments as you mentioned in your answer to Q2.

---

> > > ### Author Response · Authors · 2024-08-12
> > >
> > > Thanks for your feedback and suggestion. We will add the ablation studies mentioned in the answer of Q2 in the next version.

---

### Author Rebuttal · Authors · 2024-08-06

We sincerely thank the reviewers for their insightful feedback on our submission. We are grateful for their acknowledgment of our paper's contribution to active perception in robotic grasping. To address the questions raised regarding our method design and experimental work, we have provided comprehensive explanations aimed at clarifying the reviewers' concerns. We have also included a rebuttal PDF containing the figures referenced in our response.

We summarize the key points of our response below:

1. **The calculation of information gain:** Reviewer peVt expressed concern about the accuracy of the information gain calculation, given potential errors in rendered depth and graspness. Reviewer DbvE also mentioned possible errors in rendered depth for unseen views. In our response, we have provided a detailed illustration addressing the influence of these potential errors to view planning and included a numerical analysis in the  rebuttal PDF.
2. **Neural graspness sampling**: Reviewer 4AgP and DbvE expressed confusion about how we use neural graspness sampling for grasp poses inference. Our original paper may be unclear, leading to misunderstanding. In our response, we have provided a detailed explanation to clarify this point.
3. **Real-world experiment setting:** Reviewer 4AgP provided many valuable questions and suggestions on our real-world experiments. We have supplemented our response with additional details about the real-world scene settings and comprehensive results. Furthermore, we have included keyframe screenshots from the video of our method being executed on a real robot in the rebuttal PDF.

---

### Decision · Program_Chairs · 2024-09-25

**Decision:**

Accept (poster)

**Comment:**

The paper presents an active perception framework for robotic grasp detection, demonstrating improved performance on a large-scale benchmark and real-world validation. Reviewers appreciated the novelty but raised questions about calculating information gain, the clarity of neural grasp sampling, and the details of experimental setups. The authors responded to these concerns with additional explanations and data, satisfactorily addressing the reviewers' issues.
Considering the overall positive reviews, the authors' effective rebuttal, and the potential contributions to the field, the paper is recommended for acceptance.